# Facilitating better sharing quality of COVID-related headlines

*Irene Sophia Plank* [1*]

**1** Department of Psychiatry and Psychotherapy, LMU University Hospital, LMU Munich

⋆ irene.plank@med.uni-muenchen.de

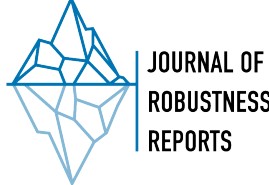

## Abstract

**Including accuracy prompts and digital literacy tips similarly decrease the likelihood to share COVID-related headlines, especially if they are false.**

## 1   Target article

A. A. Arechar, J. Allen, A. J. Berinsky, R. Cole, Z. Epstein, K. Garimella, A. Gully, J. G. Lu, R. M. Ross, M. N. Stagnaro, Y. Zhang, G. Pennycook *et al.*, *Understanding and combatting misinformation across 16 countries on six continents*, Nature Human Behaviour **7**(9), 1502 (2023), doi:10.1038/s41562-023-01641-6, Publisher: Nature Publishing Group

## 1   Goal

Arechar et al. [1] reported increased sharing quality after accuracy prompts and digital literacy tips. Given the importance of combating the spread of misinformation and their unusual analysis approach combining country-specific linear regressions and random-effect meta-analyses across countries, this robustness report investigates whether the results can be reproduced using a cumulative Bayesian linear mixed model on the full, ordinal data.

## 2   Methods

Sharing likelihood (SL) of COVID-related articles based on headlines is assessed using an online questionnaire (scale from (1) = 'moderately unlikely' to (6) = 'extremely likely'), with sharing quality (SQ) captured as the difference in SL between true and false headlines (predictor *Truth*). Data was collected in 16 countries, using translated headlines. Some participants encountered an accuracy prompt or digital literacy tips before the headlines (predictor *Condition*). Other participants were asked to rate the accuracy of the same headlines; I excluded their data as well as the data of participants who failed one of the attention tests, whose total duration was longer than the 99% percentile (94.55 min) or who chose the same

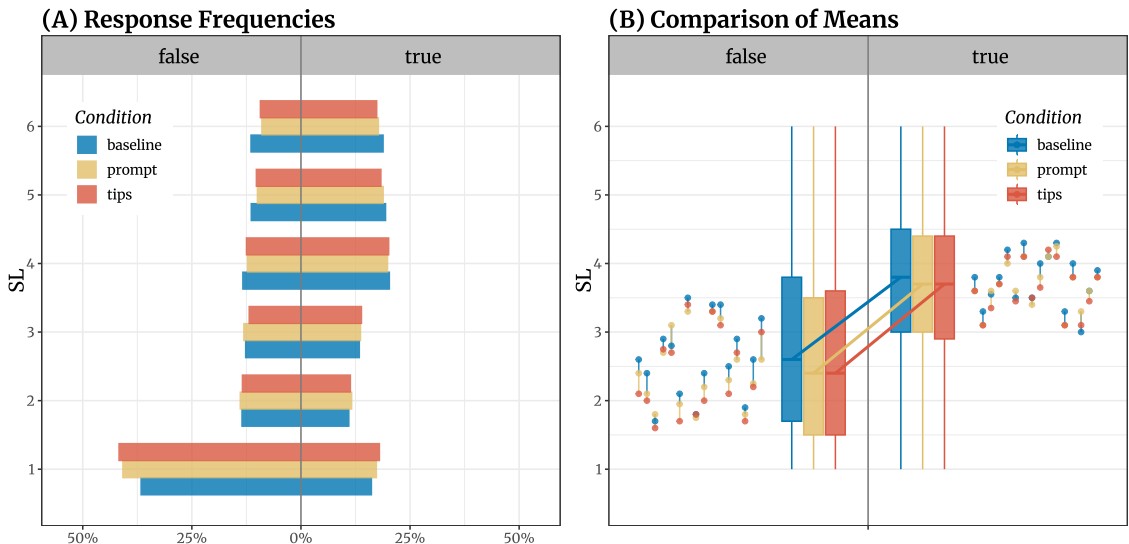

Figure 1: Subfigure (A) shows the distribution of sharing likelihood (SL) across all participants for true and false headlines in each *Condition* separately. Subfigure (B) focuses on mean SL. First, boxplots show participant-specific means of SL for true and false headlines for *Conditions* separately. Second, dots show country means for true and false headlines separately for *Conditions*. SL was lower after the prompt and the tips intervention compared to the baseline for both true and false headlines, a pattern that was observed in almost all countries. However, these effects were slightly more pronounced for false headlines indicating improved sharing quality (SQ).

rating for all headlines from this re-analysis (excluded: $n = 21,451$; included: $n = 12,835$). I modelled SL with a cumulative Bayesian linear mixed model implemented in `brms` [2]. The model included two sum-coded predictors, *Truth* (true or false headline) and *Condition* (baseline, prompt, tips), and their interaction as well as random intercepts for participant (random slopes: *Truth*), item (random slopes: *Condition*) and country (random slopes: *Truth*, *Condition* and their interaction). An alternative model included *Country* as a population-level instead of a random predictor. Estimates together with posterior probabilities of this estimate being $> 0$ are reported for each contrast.

## 3   Results

Overall, participants were more likely to share true than false headlines (true > false: *estimate* = 0.86 [0.67, 1.06], *posterior probability* = 100%). Participants who received an accuracy prompt or digital literacy tips exhibited a credibly better SQ than baseline participants (prompt > baseline: *estimate* = 0.11 [0.07, 0.16], *posterior probability* = 99.98%; tips > baseline: *estimate* = 0.1 [0.05, 0.15], *posterior probability* = 99.92%). Participants rated their SL lower after both interventions (baseline > prompt: *estimate* = 0.11 [0.07, 0.15], *posterior probability* = 100%; baseline > tips: *estimate* = 0.18 [0.12, 0.24], *posterior probability* = 99.99%), with this reduction being more pronounced for false headlines (see 1). SQ after both interventions was comparable (prompt > tips: *estimate* = 0.01 [-0.04, 0.06], *posterior probability* = 65.49%; inside *ROPE* = 100%, *HDI* = [-0.05, 0.07]). Results were similar in the alternative model; however, countries varied in the efficacy of the interventions (see S5 of the supplementary materials: https://osf.io/gpumw).

## 4  Conclusion

My results indicate that SQ was comparably improved by a preceding accuracy prompt and digital literacy tips: while people rated their overall SL lower for both true and false headlines, these effects were more pronounced for false headlines. Thus, this re-analysis supports the original claim by [1] using a cumulative Bayesian linear mixed model based on the full data.

### Acknowledgments and Disclosures

**Reproducibility**  We were able to computationally reproduce the original analysis and results.

**Code and Data Availability**  Data and R code are available on OSF: https://osf.io/7wgv2/.

**Funding**  The author received no financial support for the research, authorship, and/or publication of this article.

**Conflicts of Interest**  There are no competing or conflicting interests.

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
