# Peer review of "Facilitating better sharing quality of COVID-related headlines"

_Journal of Robustness Reports_

## Round 1 · Referee Report · Henrik Singmann (Referee 1) · 2025-9-10

Report
The target article for the robustness check was published in Nature Human Behaviour in 2023 and has already received almost 200 citations. It also uses a somewhat unusual and not always very clear analysis approach. Thus, this is a good candidate for a robustness report. The current manuscript also provides a reasonable first attempt for an alternative analysis. However, I also have several reservations with the current analysis that should be addressed before this report should be published.
The current re-analysis uses a linear mixed-effects model in a Bayesian statistical framework (using brms). Whereas this is a reasonable approach, it is currently not fully clear how this differs from the original analysis approach and what its benefits are. In a revision I hope the author can provide a short overview of the analysis approach used by the original authors and how the new approach differs from or improves the original approach. From a quick reading of the target article it was not fully clear to me to what degree individual-level data was analysed or when meta-analytic techniques were used.
My understanding is that one improvement of the present reanalysis is that country is treated as a random-effect grouping factor. However, one major downside of the current renalaysis is that the analysis is performed on by-participant aggregated data which has two negative consequences: 1. Because truth is a within-subject variable, by-participant random slopes for truth need to be included. In the present analysis this is not possible because data are aggregated on the participant-truth level (which confounds the residual error term with the truth random slope), but the original by-trial level is of course available. Without including by-participant random slopes for truth the model does not employ the maximal random effects structure justified by the design (Barr et al., 2014) which can lead to increased Type 1 errors. 2. By aggregating on the by-participant level, the item information (variable name) cannot be included in the mixed model. However, I believe it was included in the original analysis and probably should be included in the robustness analysis as well. The main reason is that if there are item effects and these are ignored in the analysis, models can have unacceptably high Type 1 error rates (Judd et al., 2012).
My suggestion for an interesting robustness reanalysis is one that includes all three possible grouping factors (participant, item, and country) and is performed on the not aggregated (i.e., trial-level data). Thus, the formula for the full model should be: value ~ Condition * Truth + (Truth | id) + (Condition * Truth | Country) + (Condition | name) Please note that given the large number of observations, a frequentist analysis (e.g., using lme4/lmerTest or the afex package) can also be perfectly acceptable if estimating such a model in a Bayesian setting is too computationally expensive. In my view, the main reason for a robustness report of the target article is not by just adding a Bayesian approach, but by exploring alternative ways of modelling the data. For example, one could compare the results of a model in which Country is treated as a random grouping factor with a model in which it is treated as a fixed factor. Does this affect the results in a meaningful way? The difference between a Bayesian and non-Bayesian analysis using the same model is unlikely to be of interest in and of itself as these typically produce the same conclusions.
I also did not fully understand why the present renalaysis only focuses on two of the target articles four conditions. If there is no specific reason for doing so I suggest including all four condition. Alternatively, provide a clear rationale for including only two conditions.
Finally, another angle that could be considered here is whether to move away from a "simple" statistical model and produce ROCs of the confidence rating data. For example, Modirrousta-Galian and Higham (2023) reanalysed a set of studies testing gamified inoculation interventions and found that these only affected the response bias and not participant's ability to distinguish true and fake news. Such an approach can be made much more complex when actually estimating say an SDT model, but this is probably not necessary. (Just to be clear, I do not think an ROC analysis is necessary for publication, this is just something to consider.)
References
Barr, D. J., Levy, R., Scheepers, C., & Tily, H. J. (2013). Random effects structure for confirmatory hypothesis testing: Keep it maximal. Journal of Memory and Language, 68(3), 255–278. https://doi.org/10.1016/j.jml.2012.11.001
Judd, C. M., Westfall, J., & Kenny, D. A. (2012). Treating stimuli as a random factor in social psychology: A new and comprehensive solution to a pervasive but largely ignored problem. Journal of Personality and Social Psychology, 103(1), 54–69. https://doi.org/10.1037/a0028347
Modirrousta-Galian, A., & Higham, P. A. (2023). Gamified inoculation interventions do not improve discrimination between true and fake news: Reanalyzing existing research with receiver operating characteristic analysis. Journal of Experimental Psychology: General, 152(9), 2411–2437. https://doi.org/10.1037/xge0001395
Recommendation
Ask for major revision
Author: Irene Sophia Plank on 2025-10-14 [id 5928]
(in reply to Report 1 by Henrik Singmann on 2025-09-10)
Dear Henrik Singmann,
Thank you for your detailed and helpful suggestions. I significantly updated the analysis strategy and the manuscript accordingly.
First, I elaborated more on the differences between the current, updated analysis strategy and the original analysis method in the Goal section, also adding the motivation behind the re-analysis.
Second, I ran a cumulative Bayesian linear mixed model with a probit link function on the full data using the formula you recommended. Additionally, I ran a second, alternative model where Country is treated as a fixed instead of a random factor. The results from both models are comparable; thus, the alternative model is presented in the supplementary materials and uploaded to OSF. The supplementary materials also include some plots regarding country-specific differences in sharing quality.
Third, I now included both interventions (tips and prompts) designed to improve sharing quality. The accuracy condition is still excluded, since participants are asked to judge accuracy and not sharing likelihood in this condition (sharing likelihood: “We are interested in the extent to which you would consider sharing them on social media if you saw them.”; accuracy: “We are interested in whether you think the information is accurate.”).
Last, while applying SDT is an interesting analysis method for fake news detection, I decided against applying it to this data, since my robustness report focuses on the sharing likelihood rather than the accuracy judgments. The sharing likelihood can capture different sharing motivations, as is shown in Figure S2 of the supplementary materials of the original article. For example, one might, correctly, infer that an article is not reliable or very manipulative, as captured in the data analysed by Modirrousta-Galian and Higham (2023), but decide to share it anyway for entertainment reasons.

Henrik Singmann on 2025-09-11 [id 5804]
The spacing in the third and fourth paragraphs is off and should look as follows (I apologise for not checking this before):
My understanding is that one improvement of the present reanalysis is that country is treated as a random-effect grouping factor. However, one major downside of the current renalaysis is that the analysis is performed on by-participant aggregated data which has two negative consequences: 1. Because truth is a within-subject variable, by-participant random slopes for truth need to be included. In the present analysis this is not possible because data are aggregated on the participant-truth level (which confounds the residual error term with the truth random slope), but the original by-trial level is of course available. Without including by-participant random slopes for truth the model does not employ the maximal random effects structure justified by the design (Barr et al., 2014) which can lead to increased Type 1 errors. 2. By aggregating on the by-participant level, the item information (variable name) cannot be included in the mixed model. However, I believe it was included in the original analysis and probably should be included in the robustness analysis as well. The main reason is that if there are item effects and these are ignored in the analysis, models can have unacceptably high Type 1 error rates (Judd et al., 2012).
My suggestion for an interesting robustness reanalysis is one that includes all three possible grouping factors (participant, item, and country) and is performed on the not aggregated (i.e., trial-level data). Thus, the formula for the full model should be: value ~ Condition * Truth + (Truth | id) + (Condition * Truth | Country) + (Condition | name) Please note that given the large number of observations, a frequentist analysis (e.g., using lme4/lmerTest or the afex package) can also be perfectly acceptable if estimating such a model in a Bayesian setting is too computationally expensive. In my view, the main reason for a robustness report of the target article is not by just adding a Bayesian approach, but by exploring alternative ways of modelling the data. For example, one could compare the results of a model in which Country is treated as a random grouping factor with a model in which it is treated as a fixed factor. Does this affect the results in a meaningful way? The difference between a Bayesian and non-Bayesian analysis using the same model is unlikely to be of interest in and of itself as these typically produce the same conclusions.

---

## Round 1 · Referee Report · Anonymous (Referee 2) · 2025-9-21

The referee discloses that the following generative AI tools have been used in the preparation of this report:
I used Claude AI to summarize the methodological approach of the original article (involving a Stata file), and used DeepL improve language. Both were used with their 21st September 2025 versions.
Report
The author conducted a robustness report of an article on misinformation on Nature Human Behavior. The original article provides only a vague description of its analytic approach, making it a useful target for a robustness report. In the robustness report, the author uses a Bayesian Gaussian linear mixed-effects model to model headline sharing. While this approach is appropriate for the data and is likely an improvement over the original article, both the rationale for the robustness analyses and the specific modeling approach could be improved before publishing the report.
- The robustness report should clearly explain why and how the robustness analysis differs from the original one and why this is important. This seems especially important given the opaque reporting of the original article. As I understand it, the original article used a two-stage random-effects meta-analytic approach. First, it clustered participants and headlines for each country separately. In the second stage, these estimates were used in a meta-analysis to pool the country-specific estimates. Using a single model for a robustness check makes sense, but this could briefly be justified. I understand that articles in this journal have relatively strict length limitations, but I think that the general summary of the original article could be shortened to make space for more justification of the robustness analysis.
- However, the robustness analysis differs from the original analysis in that it aggregates ratings within participants instead of analysing the full raw data. I don't understand why this was done, since it removes potentially important and relevant information from the data.
- There are additional analytic choices in the robustness report that are not justified, but are important: Why is only the Prompt vs. No Prompt condition investigated, instead of investigating all conditions, as in the original article? Why wasn't a random effect for the specific headline included? The specific content of headlines could be an important source of variation.
- As a minor point: The intercept is assigned a N(2.5, 1.0) prior. While this is likely unimportant for the results, a one-sentence justification for this prior could be provided in the supplement, as there is no apparent reason for this specific location choice.
In sum, I suggest conducting a robustness analysis either with the full random effects structure possible that mirrors and/or extends the original analysis, or to clearly justify a deviation from this approach (see Barr et al., 2014, for a justification of maximum random effects).
Reproducibility
The code was clear and well readable. I was able to re-run all code without error, except for the deprecated "show_guide" argument when creating Figure 1, which is an easy-to-correct issue. I could reproduce the main results reported in the robustness report. Please consider adding a license to your OSF project, so that reuse conditions of your materials are clearly spelled out.
Reference
Barr, D. J., Levy, R., Scheepers, C., & Tily, H. J. (2013). Random effects structure for confirmatory hypothesis testing: Keep it maximal. Journal of memory and language, 68(3), 255-278.
Recommendation
Ask for major revision
Dear reviewer,
Thank you for the constructive feedback. I significantly adjusted the analysis approach, now using a cumulative Bayesian linear mixed model on the full, ordinal data, including both intervention conditions (prompt and tips) as well as the baseline condition. The accuracy condition was excluded, because it asks to rate the accuracy rather than the sharing likelihood (sharing likelihood: “We are interested in the extent to which you would consider sharing them on social media if you saw them.”; accuracy: “We are interested in whether you think the information is accurate.”).
The new model includes maximum random effects for item, participant and country, but I also compared it to an alternative model with Country as a fixed effect. Since I now use a cumulative and not a Gaussian model, the priors have changed. I followed the tutorial by Solomon Kurz to set these priors: https://solomonkurz.netlify.app/blog/2021-12-29-notes-on-the-bayesian-cumulative-probit/
I elaborated on the differences and the reason behind the re-analysis in the Goal section.

---

## Round 1 · Referee Report · David Izydorczyk (Referee 3) · 2025-9-26

Report
First, and this may be due to the robustness report format or submission guidelines, I would have appreciated more background on why this particular result was selected for reanalysis from among the many reported in the original article. Was it chosen because of its substantive importance, or primarily as a showcase for the general analytical approach? A brief sketch of the original analysis, together with a rationale for the new approach using (Bayesian) linear mixed models, would also help situate the contribution more clearly.
Second, as I understand it from the report, the data were aggregated at the participant level, and the model then included a random intercept for participants and random slopes of condition and truth for countries. Given that a general recommendation for linear mixed models is to specify the maximal random-effects structure justified by the experimental design, which can help to reduce Type I errors (Baayen et al., 2008; Barr et al., 2013; Judd et al., 2012; Singmann & Kellen, 2019), I believe a better approach would have been to model the data at the participant level, including the appropriate random effects for both participants and items, if there is no reason for not doing it.
Finally, a minor point: the author refers to “included slopes” or “group-level intercepts.” Adding the conventional qualifier “random” (e.g., “random slopes,” “random intercepts”) would make it clearer.
Recommendation
Ask for minor revision
Author: Irene Sophia Plank on 2025-10-14 [id 5930]
(in reply to Report 3 by David Izydorczyk on 2025-09-26)
Dear David Izydorczyk,
Thank you for your constructive and helpful comments. The specific article was chosen due to the unusual analysis approach and the importance of finding interventions to facilitate better sharing quality to combat misinformation; I elaborated on these goals of the re-analysis in the Goal section.
I also changed the analysis approach, now using a cumulative Bayesian linear mixed model with maximal random effects on the full, unaggregated data. Furthermore, I used an alternative model with Country as a fixed rather than random effect, which produced comparable results.
Last, I adjusted the wording following your suggestions.

---

## Round 2 · Author Response

Dear editor, dear reviewers,

Thank you for the positive and detailed feedback! I detail the changes with respect to the editorial recommendation below in the list of changes. I will also upload responses to each reviewer's individual comments.

---

## Round 2 · List of Changes

1. I elaborated on the differences and the reason for the re-analysis in the Goal section.
2. I fit a cumulative model with a probit link function to the full, unaggregated data. The model includes random group-level intercepts for persons (slopes for Truth), item (slopes for Condition) and country (slopes for Condition, Truth and their interaction), as suggested by the reviewer. Additionally, I fit an alternative model with Country as a population-level predictor with comparable results.
3. I extended the model to include both interventions of sharing likelihood, i.e., the prompt and the tips condition, explaining the exclusion of the accuracy condition in the manuscript. While I still use sum contrasts, I now also compare the two intervention conditions.

---

## Round 3 · Author Response

Thank you to the editor and the reviewers for reading and assessing the revised manuscript. I addressed all remaining suggestions.

---

## Round 3 · List of Changes

• clarification of exclusion criteria for data of participant ("who chose the same rating for all headlines")
  • fixing of typo ("with a cumulative Bayesian...")
  • clarification of reporting of estimates and posterior probabilities at the end of the Methods section ("Estimates together with posterior probabilities of this estimate being > 0 are reported for each contrast.")
  • separation of the figure into two subfigures, one showing response frequencies and one showing means

---

## Editorial Decision

in_voting